# Transglutaminases Are Active in Perivascular Adipose Tissue

**DOI:** 10.3390/ijms22052649

**Published:** 2021-03-05

**Authors:** Alexis N. Orr, Janice M. Thompson, Janae M. Lyttle, Stephanie W. Watts

**Affiliations:** Department of Pharmacology and Toxicology, Michigan State University, East Lansing, MI 48824, USA; orralexi@msu.edu (A.N.O.); janthomp@msu.edu (J.M.T.); lyttlejanae@gmail.com (J.M.L.)

**Keywords:** transglutaminase, cardiovascular, perivascular adipose tissue, PVAT, TG2, FXIII

## Abstract

Transglutaminases (TGs) are crosslinking enzymes best known for their vascular remodeling in hypertension. They require calcium to form an isopeptide bond, connecting a glutamine to a protein bound lysine residue or a free amine donor such as norepinephrine (NE) or serotonin (5-HT). We discovered that perivascular adipose tissue (PVAT) contains significant amounts of these amines, making PVAT an ideal model to test interactions of amines and TGs. We hypothesized that transglutaminases are active in PVAT. Real time RT-PCR determined that Sprague Dawley rat aortic, superior mesenteric artery (SMA), and mesenteric resistance vessel (MR) PVATs express TG2 and blood coagulation Factor-XIII (FXIII) mRNA. Consistent with this, immunohistochemical analyses support that these PVATs all express TG2 and FXIII protein. The activity of TG2 and FXIII was investigated in tissue sections using substrate peptides that label active TGs when in a catalyzing calcium solution. Both TG2 and FXIII were active in rat aortic PVAT, SMAPVAT, and MRPVAT. Western blot analysis determined that the known TG inhibitor cystamine reduced incorporation of experimentally added amine donor 5-(biotinamido)pentylamine (BAP) into MRPVAT. Finally, experimentally added NE competitively inhibited incorporation of BAP into MRPVAT adipocytes. Further studies to determine the identity of amidated proteins will give insight into how these enzymes contribute to functions of PVAT and, ultimately, blood pressure.

## 1. Introduction

Transglutaminases (TGs) are a family of crosslinking enzymes that catalyze a Ca^2+^ dependent acyl-transfer. This reaction forms a covalent isopeptide bond between a γ-carboxamide group of a protein bound glutamine [1] and an amine. This amine can be a protein bound lysine [2] or a free amine donor such as the catecholamine norepinephrine (NE) or the tryptophan metabolite serotonin (5-hydroxytryptamine [5-HT]) [3]. While there are many different isoforms of TGs (TG1 through TG7, blood coagulation Factor-XIII (FXIII), and the catalytically inactive erythrocyte membrane band 4.2 [4]), only two TGs play a significant role in the vasculature. TG2, expressed in vascular endothelial and smooth muscle cells [5], is important in small artery remodeling [6] and fibroblast adhesion [7], and has been implicated in the development of cardiovascular diseases including hypertension [8] and atherosclerosis [9]. FXIII, expressed by platelets and cells of monocyte/macrophage origin [10], also has many roles in the vasculature. In addition to being essential to the blood coagulation cascade [11], FXIII promotes fibroblast adhesion [12], myocardium healing after infarction [13], and contributes to angiogenesis [14]. While these TGs have been extensively studied in the vasculature, perivascular adipose tissue (PVAT) has largely been forgotten in these studies.

PVAT is the fat the encompasses the vasculature, with the exception of vessels in the brain. This type of adipose tissue communicates extensively with the blood vessel, releasing and taking up numerous substances to influence vascular tone. In health, PVAT’s effect is largely anti-contractile [15]. However, in disease states such as obesity and hypertension, this function is diminished [16]. We have shown that PVAT contains high levels of the catecholamine NE and the tryptophan derivative 5-HT [17]. In the vasculature, NE is best known as the neurotransmitter of the sympathetic nervous system that acutely increases vascular tone through vasoconstriction. Importantly, sympathetic nervous system activity is elevated in human essential hypertension [18]. 5-HT can act as both a pressor and depressor agent [19]. It is therefore important to understand the totality of mechanisms by which 5-HT and NE, newly discovered in PVAT, use to influence blood pressure, with an ultimate goal of developing new/improved treatments for vascular diseases. Here, we determine if NE in PVAT has the additional function of serving as a substrate for transglutaminases.

The rat aorta and vena cava both contain the active transglutaminase TG2 [20], and in these vessels, TG2 incorporates NE into vascular proteins [21]. Recent studies by Penumatsa et al. showed that increased TG2 expression in lung, kidney, liver, and left ventricular tissues is associated with obesity and metabolic syndrome [22]. Other studies demonstrated that TG2 mediates ventricular [23] and aortic [24] contractility, stiffness, and function. Further, Kaartinen et al. found that human FXIIIA1 expression is increased in adipose tissue of the heavier twin in a monozygotic pair [25]. These studies highlight the importance of understanding how TGs function in adipose tissue, and more specifically in a fat intimately associated with the vasculature, PVAT.

We hypothesize that TGs are present and active in PVAT, and that free amines within PVAT serve as TG substrates. Several complementary techniques were used to test this hypothesis in tissues from the Sprague Dawley rat. RT-PCR was used to determine the subtypes of TG mRNA present in PVAT. Immunohistochemistry allowed for the visualization of TG protein through isoform specific antibodies, as well as TG activity in PVAT through use of novel TG isoform specific amine donors. Western analyses using a free amine donor, 5-(biotinamido)pentylamine (BAP), allowed for initial exploration of protein targets of TGs in PVATs and whether amines (e.g., NE) could interfere with this reaction. As they surround arteries with significantly different physiological functions and are different fat types, multiple PVATs were used, including aortic (APVAT), superior mesenteric artery (SMAPVAT), and mesenteric resistance vessel (MRPVAT). Our results support our hypothesis that TGs are present and active in PVATs.

## 2. Results

### 2.1. TG2 and FXIII mRNA Were Most Highly Expressed in PVATs

Of the 9 major TG isoforms recognized, TG2 and FXIII mRNA were the most abundantly expressed as compared to transglutaminases 1, 3, 4, 5, and 7 in APVAT, SMAPVAT, and MRPVAT (Figure 1). Due to these differences in the magnitude of subtype mRNA expression, we focused further research on TG2 and FXIII.

Two distinct primer sets were used to determine TG1 and TG3 mRNA expression levels for the purpose of validating the result of the first primer set used. mRNA amplification for the different primer sets was variable, and we can only theorize that these differences were caused by varying target locations of the primer sets, which may allow the primers to anneal more easily or completely.

### 2.2. TG2 and FXIII Proteins Were Present in PVATs

Consistent with our PCR findings, TG2 and FXIII protein were observed in sections of APVAT, SMAPVAT, and MRPVAT using brightfield immunohistochemistry (IHC) (Figure 2). Here, the media of the rat aorta and human placenta served as a positive control for TG2 and FXIII, respectively (Figure 2, upper most panel).

### 2.3. TG2 and FXIII Are Active in PVATs

Following confirmation of the presence of TG2 and FXIII protein, novel amine donors, specific for TG2 and FXIII, were used to determine enzyme activity. These amine donors were developed by Dr. Kiyotaka Hitomi, who used a phage-displayed peptide library to find the most highly reactive amino acid sequences of TG2 and FXIII. Using these amino acid sequences, the T26 and F11KA substrate peptides were synthesized to specifically identify active TG2 and FXIII, respectively [26,27]. Fresh frozen tissue sections of APVAT, SMAPVAT, and MRPVAT were incubated with TG2 (T26) and FXIII (F11KA) substrate peptides (Figure 3). Positive fluorescent labeling for TG2 and FXIII donors indicated that both TGs are active in these tissues, though not equal. Signal intensity in PVAT tissue was clearly not the magnitude of that in their respective controls of aorta and skin. However, a signal above that of the negative control tissue sections was observed. Importantly, we validated a previous finding that TG2 is active in the medial smooth muscle layers of the fresh thoracic aorta, evidenced by the brilliant signal in the medial layers of smooth muscle (TG2 positive control, Figure 3).

We next determined if TG activity could be: (1) observed by incorporation of the amine donor BAP into PVAT proteins; and (2) whether BAP incorporation into PVAT proteins could be inhibited using the TG inhibitor cystamine [28]. Utilizing the Transglutaminase Activity Assay, BAP was used as an exogenously added amine donor, tracked using a streptavidin antibody. Due to the high levels of endogenous biotin present in brown fats, such as APVAT, we focused our attention on the white fat MRPVAT (Appendix A). 

Figure 4A demonstrates incorporation of BAP into many proteins of MRPVAT, observed as stronger signal in lanes where only CaCl_2_ was present. Importantly, the isoform non-specific inhibitor cystamine significantly reduced the calcium-driven TG reaction, visualized with reduced signal in lanes where both CaCl_2_ and cystamine were present (Figure 4A). Cystamine alone lanes act as a negative control, proving that cystamine is reducing BAP incorporation in a calcium-dependent manner. While it would be assumed that cystamine alone lanes should appear blank, there is still endogenous calcium within the protein homogenate that likely drives the TG reaction, as well as low concentrations of endogenous biotin that may be tagged by the streptavidin secondary. These data are quantified in Figure 4B, in which all bands in each lane are included in the scatter plot. The ability of cystamine to inhibit the incorporation of the biotin labeled amine suggests that BAP incorporation into proteins of MRPVAT is TG-dependent. 

### 2.4. Adipocytes Possess the Majority of TG Activity

Aware that PVAT contains many cells in addition to adipocytes, we repeated the experiment shared in Figure 4 using adipocyte and stromal vascular fraction (SVF) protein isolated from MRPVAT. BAP was incorporated into protein from adipocytes to a greater extent and magnitude than SVF derived protein, visualized by a stronger signal in lanes where CaCl_2_ alone was present (Figure 5A). Moreover, BAP incorporation into adipocyte, but not SVF, protein was significantly reduced by cystamine, observed in lanes where both CaCl_2_ and cystamine were present (Figure 5A and quantified in Figure 5B). No significant differences in signal between any conditions for SVF protein samples were observed, indicating that TGs are less active in SVF than adipocytes. Lanes containing only cystamine again acted as a negative control to ensure the reduction of BAP incorporation was calcium dependent. 

### 2.5. Catecholamine NE Competitevly Reduces BAP Incorporation in Adipocyte Homogenates

To determine whether amine donors, present in adipocytes, have the potential to act as a substrate for TGs by interfering with BAP incorporation into MRPVAT adipocyte derived proteins, the catecholamine NE was included in the Transglutaminase Activity Assay reaction solution in increasing concentrations. BAP incorporation was significantly reduced when 20 mM and 40 mM NE was added to the reaction solution, as similarly observed with cystamine (Figure 6A). Lanes containing cystamine acted as a control to compare the reduction of BAP incorporation between trials where NE was and was not present (Figure 6A). These data are quantified in Figure 6B; all bands in each lane are included in this quantification. This study supports that NE can serve as a substrate for PVAT transglutaminases. 

## 3. Discussion

These experiments were conducted to determine if TGs are present in PVAT, and if so, whether TGs are functionally active. TG2 and FXIII mRNA and protein were both present and active in all PVATs studied, and functional in MRPVAT. Functionality was assessed first by incorporation of an isoform specific amine donor in fresh frozen tissue sections, which bind to highly reactive glutamine donors of TG2 [26] and FXIII [27], and then by BAP incorporation into protein homogenate of PVAT and adipocytes/SVF that was sensitive to cystamine inhibition. Importantly, separation of adipocytes and the SVF showed TGs are more active in adipocytes, rather than the SVF. Finally, we observed a competitive inhibition of BAP incorporation by the catecholamine NE in isolated MRPVAT adipocytes, consistent with the idea that TGs use NE as a substrate in these cells. With these data, we support our hypothesis and maintain that TGs are present and active in PVAT. 

To our knowledge, this is the first time TG2 and FXIII have been found to be present and active in PVAT. Due to PVAT’s close proximity to the blood vessel and its ability to influence vascular tone, the activity of TGs in PVAT may be relevant to the regulation of a myriad of functions within the vasculature. For example, overexpression of TGs in PVAT may lead to PVAT dysfunction or stiffening, and therefore impact vascular tone and stiffness. 

The ability for TGs to use NE as a substrate and amidate PVAT proteins may have great physiological consequences as well. TGs can facilitate incorporation of NE into vascular (tunica intima, media and adventitial) proteins [21]. Here, we show that the proteins of PVAT may also be able to incorporate NE, a substrate that is stored in PVATs. In our experimental model, NE is likely acting in competition with BAP as a substrate for TGs. The success of this experiment supports that BAP is handled by TGs in a manner similar to NE, confirming that our experimental model truly examines the transamidation properties of TGs. Additionally, this discovery further challenges the belief that NE only elicits a physiological response through adrenergic receptors, thereby increasing the known physiological effects that NE has. 

There are several limitations to note from our present study. Only TG2 and FXIII were investigated for activity in PVAT, though the mRNA of several other TGs were found to be present. Our conclusion does not mean that these other TGs are inactive, but rather TG2 and FXIII are the most present TG isoforms in PVAT and that they are active. The use of BAP as a comparator for NE as a TG substrate assumes that NE and BAP are handled in a similar manner. This competition, rather than a direct labelling of proteins by NE, was done due to lack of availability of a labelled NE molecule. We have used NE-biotin in the past [21] but concerns were raised for (1) the dependence of movement of this molecule by a biotin transporter vs. an amine transporter to enter the system; and (2) availability of the appropriate amine in the NE molecule that would allow it to participate in the TG reaction. As such, we have used this indirect approach to test the idea that catecholamines, such as NE, which are abundant in PVAT adipocytes, serve as TG substrates. That possibility helps to potentially answer the question of the function of amines observed in adipocytes. 

Further limitations include the use of the TG inhibitor cystamine, which has some non-specific effects [29]. Finally, the mesenteric resistance adipose tissue was the only fat used for the Transglutaminase Activity Assay. This PVAT was the only one that we were able to visualize BAP incorporation into due to the low levels of endogenous biotin present within MRPVAT. As seen in Appendix A, the brown fats of brown fat pad and APVAT had stronger endogenous biotin staining than the mixed fat SMAPVAT. Both brown and mixed fats had stronger staining than the white fats of retroperitoneal fat and MRPVAT, which both had similar biotin content to aorta. We therefore cannot conclude whether TGs use amines as substrates in other PVATs. This is an important point because brown fats, such as the scapular fat pad and APVAT, contain significantly more catecholamines/mg tissue than do white fats. NE content in APVAT is ~730 ng/g tissue, while in MRPVAT it is ~96 ng/g tissue [30]. Our studies of MPVAT, containing lower amounts of catecholamines, could underestimate the power of the interaction of NE and TGs in PVAT. 

Future studies should focus on the characterization of protein targets that TGs amidate within PVAT, and how their function may change with the addition of amines to them. It is clear that not all proteins are amidated by TGs. In Figure 4, Figure 5 and Figure 6, some protein bands are more significantly incorporating BAP than others and thus this is not a global action. By determining which proteins TGs are acting upon, more insight may be given into how protein amidation affects various processes within PVAT, and more specifically adipocytes. Additionally, researching whether TGs functionally amidate proteins in APVAT and SMAPVAT will be important in understanding how the PVAT of these vessels contribute to vascular tone. 

This study has important implications. First, it defines a new location of functional TGs. Second, this study presents a novel mechanism that catecholamines, specifically NE, may serve as a substrate for TGs in PVAT, further challenging the belief that catecholamines require a receptor to elicit a physiological response. Finally, this work expands on the knowledge of PVAT’s capabilities as a whole, providing insight into different ways it may regulate blood pressure.

## 4. Materials and Methods 

### 4.1. Animal Use

Normal male 200–250 g Sprague Dawley rats (Charles River, Indianapolis, IN, USA) were used in accordance with the Guide for the Care and Use of Laboratory Animals (8th edition, 2011). Rats were anesthetized with pentobarbital (80 mg/kg intraperitoneal injection), exsanguinated, and tissues were removed. Fat surrounding the vessels of interest (aorta, superior mesenteric artery, and mesenteric resistance vessels) were considered their respective PVATs. PVAT was removed from vessels in Krebs-Ringer bicarbonate buffer (KRBB; 135 mmol/L NaCl, 5 mmol/L KCl, 1 mmol/L MgSO_4_, 0.4 mmol/L K_2_HPO_4_, 5.5 mmol/L glucose, 20 mmol/L HEPES, 10 mL antibiotic/antimycotic, pH to 7.4) using a stereomicroscope and micro scissors. Retroperitoneal fat was removed from the retroperitoneal cavity behind the kidneys and brown fat pad was removed from the subscapular region for use in Western blot analysis to determine endogenous biotin content (Appendix B). All protocols follow the Michigan State University Institutional Animal Care and Use Committee (IACUC) guidelines (Protocol # 02–18-026, approved 18 February 2018).

### 4.2. Adipocyte and SVF Isolation

Dissected MRPVAT was placed in sterile 1.5 mL tubes with 1 mg of collagenase type I (LS004196, Worthington Biochemical, Lakewood, NJ, USA) and 4% BSA (A2153, Sigma, Burlington, MA, USA) in KRBB. PVAT strips were minced using small scissors and placed in a 37 °C rotating incubator for 45 min, or until the tissue was digested into an opaque solution and no large PVAT fragments were visible. Once digestion was complete tubes were centrifuged at 200× *g* for 5 min at room temperature. Using a sterile 23 gauge needle, the subnatant was discarded and SVF pellet collected. Eight hundred (800) μL of KRBB was then added to each tube of adipocytes and tubes were gently finger flicked to mix. Tubes were centrifuged again at 200× *g* at room temperature for 5 min and subnatant was discarded. This process was repeated for a total of 3 washes and adipocytes were collected for protein quantification.

After the SVF pellet was isolated from the adipocytes, the SVF was centrifuged at 1500× *g* for 5 min in a new sterile tube and the supernatant was removed. The pellet was resuspended in 50 μL of 1× diluted 10× Red Blood Cell Lysis Buffer (420301, BioLegend, San Diego, CA, USA) and placed in a 37 °C rotating incubator for 1 min. The reaction was stopped by adding 200 μL KRBB into the tube. The sample was spun down again, and supernatant removed for a total of 3 washes. After washing, the pellet was collected for protein quantification. 

### 4.3. Protein Isolation

PVAT protein was isolated using Omni Bead Ruptor tubes and PBS with protease inhibitors (10 mg/mL aprotinin/leupeptin, 100 mM PMSF, and 10 mM orthovanadate) at a 1:1 volume of fat. Fat was homogenized using the Omni Bead Ruptor (Omni, Inc., Jennesaw, GA, USA). Triton X-100 was incorporated into the PVAT homogenate at a 1% concentration and the homogenate was frozen overnight. Homogenate was removed from tubes and centrifuged, and the clear supernatant was collected and analyzed for protein concentration using a Bicinchoninic Acid Assay (BCA1-1KT, Sigma, St. Louis, MO, USA). 

For adipocyte protein isolation, samples were frozen at −80 °C in PBS with protease inhibitors at a 1:1 volume of fat and 1% Triton X-100 overnight. The use of the Omni Bead Ruptor was unnecessary to lyse the cells. 

### 4.4. RT-PCR

PVAT samples were homogenized using the Omni Bead Ruptor (Omni, Inc.) and RNA Lysis Buffer (R1060, Zymo Research, Irving, CA, USA). RNA was isolated from the homogenate using the Zymo Quick-RNA Mini Prep Kit (R1055, Zymo Research), then quantified on a Nanodrop 2000c spectrophotometer (Thermo Fisher Scientific, Waltham, MA, USA, RRID: SCR_018042). cDNA was reverse transcribed using the High Capacity cDNA Reverse Transcription Kit (4368814, Thermo Fischer Scientific). RT-PCR was performed on an Applied Biosystems 7500 FAST Real-Time PCR System (Thermo Fischer Scientific) using Fast SYBR Green Master Mix (4385612, Thermo Fischer Scientific), utilizing cycle parameters of 95 °C for 10 min, 95 °C for 15 s, and 60 °C for 1 min for 40 cycles, followed by a melt curve to determine the presence of a single PCR product. Measures were compared by running the housekeeping gene *β-2-microglobulin* (B2m). Primers were obtained from RealTimePrimers.com (Elkins Park, PA, USA). 

TG1 (item # VRPS-6259)

Set 1

Forward Primer 5′-GAC TAC TCT CGA GGC ACC AA-3′

Reverse Primer 5′-CGT GTG CAG AGT TGA AGT TG-3′

Set 2

Forward Primer 5′-CAT CCT CTT CAA TCC CTG GT-3′

Reverse Primer 5′-TCA AAC TGG CCA TAA TTC CA-3′

TG2 (item # VRPS-6262)

Forward Primer 5′-GGG AAT ACG TCC TCA CAC AG-3′

Reverse Primer 5′-GTC ATC ATT GCA GTT GAC CA-3′

TG3 (item # VRPS-6261)

Set 1

Forward Primer 5′-GAA CCT GGA ACG GTA GTG TG-3′

Reverse Primer 5′-GCT ATC ACT GCC TTT CTC CA-3′

Set 2

Forward Primer 5′-TAC TAT GAC GCC ATG GGA AA-3′

Reverse Primer 5′-AAG TTC TGG TCC ACC TCA CC-3′

TG4 (item # VRPS-6262)

Forward Primer 5′-ATA GAA TGC ACC CCA GTG AA-3′

Reverse Primer 5′-ACA TGC TTA CCA AGG CTC AG-3′

TG5 (item # VRPS-6263)

Forward Primer 5′-TAT TTT CAA ACC CCC TCT CG-3′

Reverse Primer 5′-TCT GCC TTT GTC CAC TCT TG-3′

TG7 (item # VRPS-6264)

Forward Primer 5′-GGA CAG CCT GTG AAA TAT GG-3′

Reverse Primer 5′-GGT GGA AGG TCT TTC CTG AT-3′

FXIII (item # VRPS-1953)

Forward Primer 5′-AAA CTG CCC TGA TGT ATG GA-3′

Reverse Primer 5′-CCC CAG TGT AGA AGG TGA TG-3′

B2m (item # VRPS-553)

Forward Primer 5′-TGC TAC GTG TCT CAG TTC CA-3′

Reverse Primer 5′-GCT CCT TCA GAG TGA CGT GT-3′

Data is reported as cycle threshold values of 40 cycles and normalized to β-2-microglobulin expression.

### 4.5. Brightfield Immunohistochemistry

Tissue sections of aorta, superior mesenteric artery, and mesenteric resistance vessels with their respective PVATs from normal male Sprague Dawley rats were paraffin-embedded and slides made. All sections were incubated for one hour with a species-specific blocking serum (goat anti-rabbit blocking kit, PK-6106, Vector Laboratories, Burlingame, CA, USA) to prevent non-specific primary antibody binding. Primary antibody for TG2 (ab421, Abcam, Cambridge, MA, USA, RRID: AB_304372) and FXIII (ab83895, Abcam, RRID: AB_1860464) were applied to experimental sections at 1:25 and 1:50 concentrations, respectively overnight at 4 °C, while control sections continued to incubate in blocking serum alone. After washing both sections of the slide the following day 3 times with PBS, secondary antibody from the same species specific kit was applied for 30 min at room temperature, followed by a 30 min incubation with the ABC Elite Kit (PK-6100, Abcam) at room temperature. Slides were then developed with 3,3-diaminobenzidine (DAB Peroxidase Substrate Kit, SK-4100, Vector), counterstained with Hematoxylin QS (H-3404, Vector), and imaged on a Nikon Digital Sight DS-Qi1 camera using mmi Cell Tools software (MMI, Eching, Germany). Paraffin embedded rat aorta and human placenta were used as a positive control for TG2 and FXIII, respectively. 

### 4.6. In Situ TG Activity Assay

Aorta, superior mesenteric artery, and mesenteric resistance vessels with their respective PVATs were harvested from normal male Sprague Dawley rats, flash frozen and sectioned for fresh, unfixed samples. After allowing slides to equilibrate to room temperature for 45 min, sections were blocked with 1% BSA and 150 mM NaCl in PBS for 15 min, followed by incubation with Avidin/Biotin Blocking Kit (SP-2001, Vector), all at room temperature. Experimental sections were then incubated with reaction solution (100 mM CaCl_2_, 1 M Tris-Cl pH 8.0, 20 mM DTT, and 0.1 mM TG2 (B008, Zedira, Darmstadt, Germany) or 0.02 mM FXIII (B010, Zedira) substrate peptide) for 90 min at 37 °C. Stop solution (25 mM EDTA in PBS) was applied to all sections for 5 min at room temperature. All sections were then incubated with Rhodamine Red-X-conjugated Streptavidin secondary (016-290-084, Jackson Immuno Research Laboratories, Inc., West Grove, PA, USA, RRID: AB_2337247) for 60 min at 37 °C. ProLong Gold Antifade Reagent with DAPI (P36931, Invitrogen, Carlsbad, CA, USA) was applied to all sections, and slides were imaged using a Nikon Eclipse inverted microscope equipped with a Nikon Digital Sight DS-Qi1 camera, using mmi Cell Tools Software (MMI, Eching, Germany). Fresh frozen sections of rat aorta or rat skin [31] were used as a positive control for TG2 and FXIII, respectively. 

### 4.7. Transglutaminase Activity Assay

PVAT protein was isolated using the protocol described. For each assay, 50 μg of protein was incubated with a reaction solution containing 4 mM 5-(biotinamido)pentylamine (BAP; EZ-Link Pentylamine-Biotin, 21345, Thermo Scientific), either 5 mM CaCl_2_, 2 mM cystamine, or both, and TG buffer (50 mmol/L Tris-Cl, 149.89 mmol/L NaCl, pH 7.5) for 60 min at 37 °C. The reaction was stopped by adding an equal volume of 2× SDS buffer with β-mercaptoethanol and boiling for 10 min. Samples were separated by SDS-PAGE and transferred onto nitrocellulose membranes. Total protein transferred to the membrane was determined using REVERT Total Protein Stain (926-11021, LI-COR Biosciences, Lincoln, NE, USA) and imaged on the 700 channel using the LI-COR Odyssey CLx (LI-COR Bioscience, RRID: SCR_014579). After removing the total protein stain using REVERT Reversal Solution (926-11013, LI-COR Bioscience), the membrane was incubated for three hours at 4 °C with 10 mL blocking buffer made from 4% chick egg ovalbumin and 0.1 mL 2.5% sodium azide in TBS-T. The membrane was washed 3 times with TBS-T, and then incubated with IRDye 800CW Streptavidin secondary (926-32230, LI-COR Biosciences) in blocking buffer at 4 °C for 1 h to detect biotin labeled proteins. The membrane was imaged again using the LI-COR Odyssey CLx on the 800 channel and quantified using Image Studio (LI-COR Biosciences, RRID: SCR_015795). 

For Transglutaminase Activity Assay experiments comparing the effect of catecholamines on BAP incorporation, 4 mM, 20 mM, 40 mM NE (A7256, Sigma), or cystamine was added to the reaction solution prior to the 60 min incubation.

### 4.8. Data Analysis

All values reported represent means ± SEM for a number of biological replicates, denoted by *n*. Brightness and contrast were equally adjusted across the whole image for clarity in all IHC and in situ TG Activity Assay image sets, never in part. Exposure times and lookup tables were set equally for all sections in an experimental set. The images of the whole blots are shared directly in the figures of this paper. Relative signal in Transglutaminase Activity Assay experiments was determined using Image Studio software. Total signal in each lane of the Streptavidin blot depicting BAP incorporation was divided by the total signal in the Total Protein blot of the same lane to obtain a percentage of signal. Brightness and contrast were adjusted on Western images (whole, not in part) for clarity, but these adjustments do not affect measurements in the Image Studio software. Statistical analysis was performed using two-way ANOVA with a Bonferroni’s correction for multiple comparisons. Data were graphed using Graph Pad Prism (GraphPad Software, San Diego, CA, USA, RRID: SCR_002798). 

## Figures and Tables

**Figure 1 ijms-22-02649-f001:**
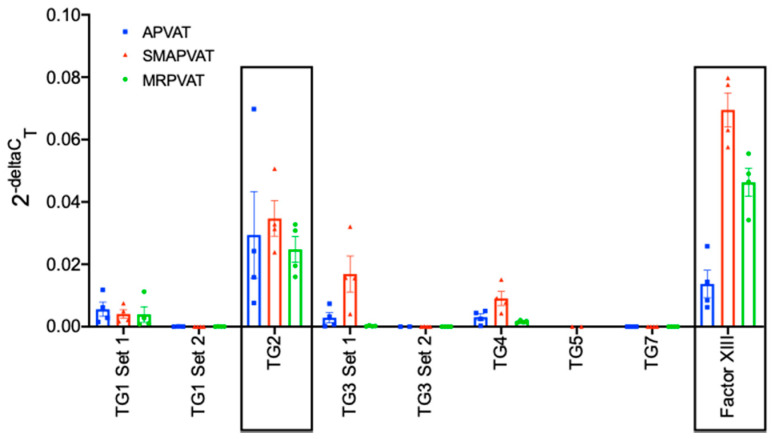
Expression of transglutaminase family genes in APVAT (blue), SMAPVAT (red), and MRPVAT (green). Multiple primers were used for TG1 and TG3, as indicated by set 1 and set 2. Boxes indicate target transglutaminases used for further studies. β-2-microglobulin was used as the reference gene. Data are expressed as mean ± SEM. *n* = 4 rats for each group.

**Figure 2 ijms-22-02649-f002:**
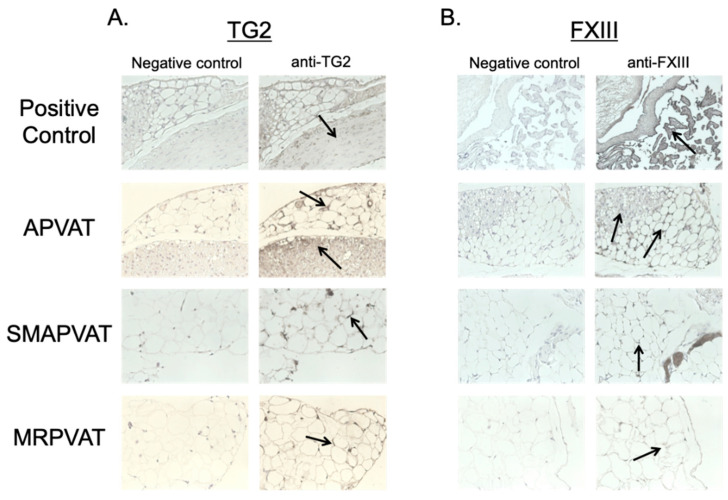
Brightfield immunohistochemistry images using (**A**) TG2 or (**B**) FXIII primary antibodies to determine location of TGs in APVAT, SMAPVAT, and MRPVAT. Rat aorta and human placenta were used as positive controls for TG2 and FXIII, respectively. Arrows indicate locations of positive TG staining. Images are representative of *n* = 4.

**Figure 3 ijms-22-02649-f003:**
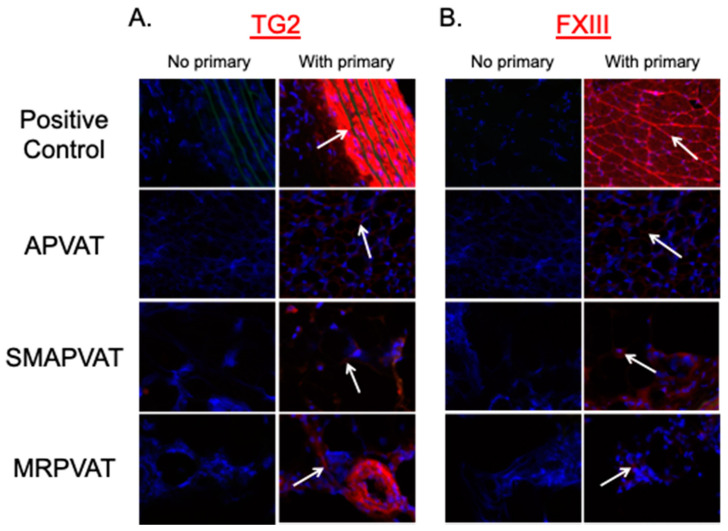
Images of in situ activity signal using (**A**) TG2 T26 or (**B**) FXIII F11KA substrate peptides to determine TG activity in fresh frozen tissue sections. Red staining and arrows indicate locations of TG activity. Rat aorta and skin were used as positive controls for TG2 and FXIII, respectively. Images are representative *n* ≥ 4.

**Figure 4 ijms-22-02649-f004:**
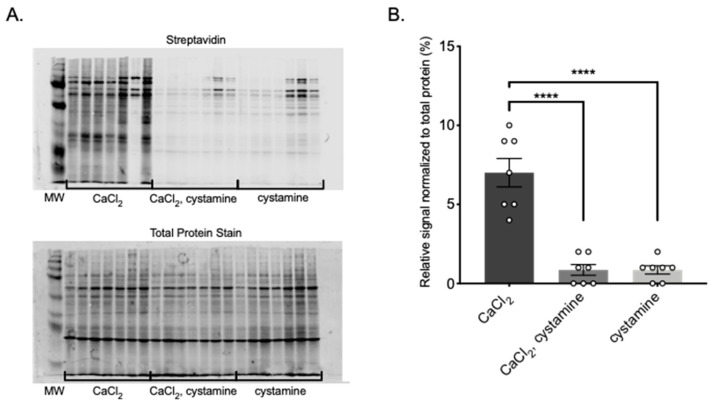
BAP incorporation into MRPVAT, visualized with IRDye 800CW Streptavidin secondary (**A** top) and total protein stain (**A** bottom). Quantification of the relative signal normalized to the total protein stain (**B**). Data are expressed as mean ± SEM. *n* = 7 rats per group. **** *p* < 0.0001 compared to only CaCl_2_ treated MRPVAT protein by one-way ANOVA with Bonferroni’s multiple comparison.

**Figure 5 ijms-22-02649-f005:**
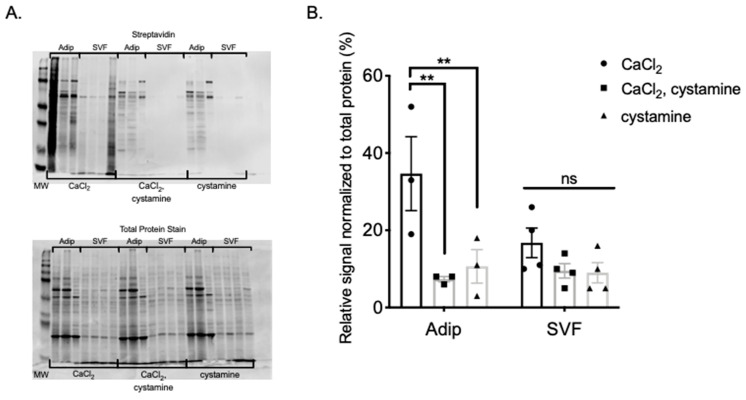
BAP incorporation into MRPVAT adipocytes and SVF visualized with IRDye 800CW Streptavidin secondary (**A** top) and total protein stain (**A** bottom). Quantification of the relative signal normalized to the total protein stain (**B**). Data are expressed as mean ± SEM. *n* = 3 rats per group for adipocyte samples and *n* = 4 rats per group for SVF samples. ** *p* < 0.01 compared to only CaCl_2_ treated MRPVAT adipocyte protein by one-way ANOVA with Bonferroni’s multiple comparison.

**Figure 6 ijms-22-02649-f006:**
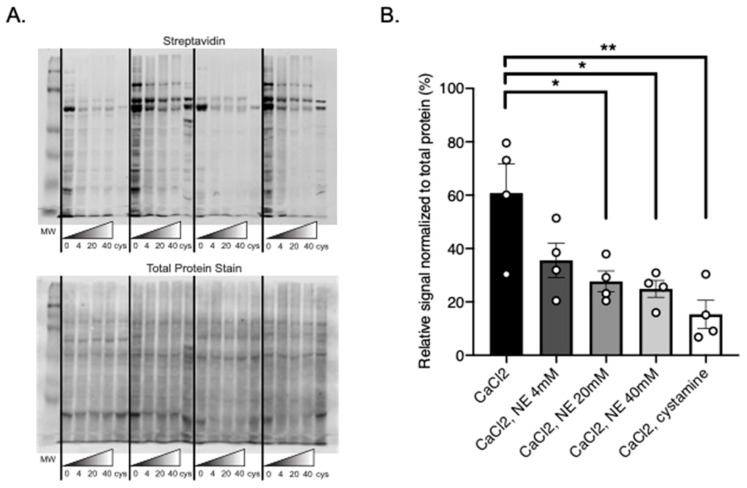
BAP incorporation into MRPVAT adipocytes in the presence of increasing NE concentrations (mM), denoted by numbers below gradient triangles (**A**), visualized with IRDye 800CW Streptavidin secondary (**A** top) and total protein stain (**A** bottom). Quantification of the relative signal normalized to the total protein stain (**B**). Data are expressed as mean ± SEM. *n* = 4 rats per group. ** *p* < 0.01, * *p* < 0.05 compared to only CaCl_2_ treated MRPVAT adipocyte protein by one-way ANOVA with Bonferroni’s multiple comparison.

## Data Availability

Data will be available upon request.

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
