# Peer review of "Transglutaminases Are Active in Perivascular Adipose Tissue"

_ijms, 2021, doi:10.3390/ijms22052649_

Round 1

Reviewer 1 Report

In the article Alexis N. Orr et al. the authors have demonstrated that transglutaminase enzymes (TGs) are present and active in perivascular adipose tissue (PVATs). The work, even if still preliminary, propose an interesting approach to investigate TGs activity in tissue compartment, focusing the attention on TG2 and FXIII expression and activity in a specific tissue.

Revisions:

Line 69-79: the authors have explained the aim of work and the approach used for their investigate but are not very clear to me, for example line 69-70 is confusing; please organize the text better.

Result 2.1: The authors used multiple primers for TG1 and TG3 and different primers set showed different level of amplification, please discuss in the result this point.

Figure 3: the image showed in figure 3 give a qualitative information, please associate a quantitative date of signal of TG2 and FXIII substrate. Usually immunofluorescence signals can be measured with microscopes or flow cytometers.

Discussion, line 182-183:  the expression “TG2 and FXIII are the most active isoforms in PVAT” is not correct because the authors have measured level of expression of different TGs by RT-PCR and the activity only for TG2 and FXIII. So the authors need to be clearer about what they mean.

Reviewer 2 Report

The manuscript is suitable for publication.

Author Response

Thank you.

Reviewer 3 Report

The authors investigated the expressions and activities of transglutaminases, protein cross-linking enzyme, in the perivascular tissues (PVAT), by immunohistochemical analysis and in situ activities. They also clarified that monoamine (HE) contained in this tissue was possible to be substrate. Based on these results, they proposed contribution of transglutaminases to possible function of PVAT.

Most experiments were carried out straightforward and interesting data have been demonstrated. However, several unclarified points were raised to be addressed. Also several descriptions and preparation of the figures in the manuscript should be revised.

Major points

  1. What are possible substrates that incorporate BPA and/or to react NE ? If there is speculation or information, this would further clarify the function of the enzyme.
  2. Why can TG2 and FXIII activities be distinguished using these substrate peptides (Fig. 3)? They should explain or quote the article.
  3. In Fig. 3, they used skin as a positive control expressing FXIII activity. Does skin express FXIII (in spite of blood transglutaminase)?

Minor points

  1. In Fig. 2, description above the pictures “No primary” and “With primary” could be changed to more appropriately, such as “(negative) control” and “anti-TG2 or anti-FXIII”.
  2. line 108: In the legend to figure 3, the description “immunofluorescent” is incorrect. “in situ activity signal”.

6. In Figure, 6, one picture may be enough as representative one showing most clear signal with enlarged size. 

Round 2

Reviewer 3 Report

The first manuscript has been well improved. The study is valuable to show the novel functions of transglutaminase reactions. This is worth for publication.